# Bayes Consistency vs. $\mathcal{H}$-Consistency:
# The Interplay between Surrogate Loss Functions and the Scoring Function Class

**Mingyuan Zhang**
University of Pennsylvania
Philadelphia, PA 19104
myz@seas.upenn.edu

**Shivani Agarwal**
University of Pennsylvania
Philadelphia, PA 19104
ashivani@seas.upenn.edu

## Abstract

A fundamental question in multiclass classification concerns understanding the consistency properties of surrogate risk minimization algorithms, which minimize a (often convex) surrogate to the multiclass 0-1 loss. In particular, the framework of calibrated surrogates has played an important role in analyzing *Bayes consistency* of such algorithms, i.e. in studying convergence to a Bayes optimal classifier (Zhang, 2004; Tewari and Bartlett, 2007). However, follow-up work has suggested this framework can be of limited value when studying $\mathcal{H}$-*consistency*; in particular, concerns have been raised that even when the data comes from an underlying linear model, minimizing certain convex calibrated surrogates over linear scoring functions fails to recover the true model (Long and Servedio, 2013). In this paper, we investigate this apparent conundrum. We find that while some calibrated surrogates can indeed fail to provide $\mathcal{H}$-consistency when minimized over a natural-looking but naïvely chosen scoring function class $\mathcal{F}$, the situation can potentially be remedied by minimizing them over a more carefully chosen class of scoring functions $\mathcal{F}$. In particular, for the popular one-vs-all hinge and logistic surrogates, both of which are calibrated (and therefore provide Bayes consistency) under realizable models, but were previously shown to pose problems for realizable $\mathcal{H}$-consistency, we derive a form of scoring function class $\mathcal{F}$ that enables $\mathcal{H}$-consistency. When $\mathcal{H}$ is the class of linear models, the class $\mathcal{F}$ consists of certain piecewise linear scoring functions that are characterized by the same number of parameters as in the linear case, and minimization over which can be performed using an adaptation of the min-pooling idea from neural network training. Our experiments confirm that the one-vs-all surrogates, when trained over this class of *nonlinear* scoring functions $\mathcal{F}$, yield better *linear* multiclass classifiers than when trained over standard linear scoring functions.

## 1 Introduction and Background

Consider a standard multiclass classification problem, with instance space $\mathcal{X} \subseteq \mathbb{R}^d$, label space $\mathcal{Y} = [n] := \{1, \ldots, n\}$ with $n > 2$ classes, and standard 0-1 loss $\ell_{0\text{-}1} : \mathcal{Y} \times \mathcal{Y} \rightarrow \mathbb{R}_+$ given by $\ell_{0\text{-}1}(y, \widehat{y}) = \mathbf{1}(\widehat{y} \neq y)$. There is an unknown probability distribution $D$ on $\mathcal{X} \times \mathcal{Y}$; given a training sample $S = ((\mathbf{x}_1, y_1), \ldots, (\mathbf{x}_m, y_m))$ containing examples drawn i.i.d. from $D$, the goal is to learn a classifier $h : \mathcal{X} \rightarrow \mathcal{Y}$ with small 0-1 generalization error on new examples drawn from $D$:

$$\text{er}_D^{0\text{-}1}[h] \;=\; \mathbf{E}_{(X,Y)\sim D}\big[\ell_{0\text{-}1}(Y, h(X))\big] \;=\; \mathbf{P}_{(X,Y)\sim D}\big(h(X) \neq Y\big). \tag{1}$$

A *Bayes consistent* algorithm is one which, given enough training examples, learns a classifier whose generalization error approaches the *Bayes optimal error*:

$$\text{er}_D^{0\text{-}1,*} \;=\; \inf_{h:\mathcal{X}\rightarrow\mathcal{Y}} \text{er}_D^{0\text{-}1}[h]. \tag{2}$$

Table 1: Examples of convex surrogate losses used by various multiclass classification algorithms, together with a summary of some previous consistency results (here $z_+ = \max(0, z)$). In this paper, we show that one-vs-all surrogates can in fact achieve $\mathcal{H}$-consistency if minimized over the right scoring function class.

| Algorithm | Surrogate loss $\psi : \mathcal{Y} \times \mathbb{R}^n \to \mathbb{R}_+$ | Universally calibrated? | Realizable calibrated? | Realizable $\mathcal{H}_{\text{lin}}$-consistent? |
|---|---|---|---|---|
| Multiclass logistic regression | $\psi_{\text{mlog}}(y, \mathbf{u}) = -u_y + \ln\left(\sum_{y'=1}^n \exp(u_{y'})\right)$ | ✓ | ✓ | ✓ |
| Crammer-Singer multiclass SVM | $\psi_{\text{CS}}(y, \mathbf{u}) = \max_{y' \neq y}(1 - (u_y - u_{y'}))_+$ | ✕ | ✓ | ✓ |
| One-vs-all logistic regression | $\psi_{\text{OvA,log}}(y, \mathbf{u}) = \ln(1 + e^{-u_y}) + \sum_{y' \neq y} \ln(1 + e^{u_{y'}})$ | ✓ | ✓ | ✕ (we give a fix) |
| One-vs-all SVM | $\psi_{\text{OvA,hinge}}(y, \mathbf{u}) = (1 - u_y)_+ + \sum_{y' \neq y}(1 + u_{y'})_+$ | ✕ | ✓ | ✕ (we give a fix) |

On the other hand, for a class of models $\mathcal{H} \subset \{h : \mathcal{X} \to \mathcal{Y}\}$, an $\mathcal{H}$-*consistent* algorithm is one which, given enough training examples, learns a classifier whose generalization error approaches the *optimal error in $\mathcal{H}$*:

$$\text{er}_D^{\text{0-1}}[\mathcal{H}] = \inf_{h \in \mathcal{H}} \text{er}_D^{\text{0-1}}[h]. \tag{3}$$

Since minimizing the discrete 0-1 loss directly is generally computationally hard, a popular approach to multiclass classification is to learn $n$ real-valued *scoring functions* $f_1, \ldots, f_n : \mathcal{X} \to \mathbb{R}$, one for each class, by minimizing a (often convex) surrogate loss, and then given a new test point $\mathbf{x} \in \mathcal{X}$, to predict a class $y$ with highest score $f_y(\mathbf{x})$. Specifically, given a training sample $S$ as above, a surrogate loss $\psi : \mathcal{Y} \times \mathbb{R}^n \to \mathbb{R}_+$, and a *scoring function class* $\mathcal{F} \subset \{\mathbf{f} : \mathcal{X} \to \mathbb{R}^n\}$, a $(\psi, \mathcal{F})$ *surrogate risk minimization algorithm* finds a vector of $n$ scoring functions $\widehat{\mathbf{f}} : \mathcal{X} \to \mathbb{R}^n$ by solving

$$\widehat{\mathbf{f}} \in \text{argmin}_{\mathbf{f} \in \mathcal{F}} \frac{1}{m} \sum_{i=1}^m \psi(y_i, \mathbf{f}(\mathbf{x}_i)), \tag{4}$$

and then returns a classifier $\widehat{h} : \mathcal{X} \to \mathcal{Y}$ given by

$$\widehat{h}(\mathbf{x}) \in \text{argmax}_{y \in [n]} \widehat{f}_y(\mathbf{x}). \tag{5}$$

This approach includes several popular multiclass learning algorithms, such as multiclass logistic regression, various forms of multiclass SVMs [15, 4, 6, 5], one-vs-all logistic regression, and one-vs-all SVM; see Table 1 for a summary of the surrogate losses used by some of these algorithms.

A natural question then is: Under what conditions do such surrogate risk minimization algorithms provide Bayes consistency or, for various classes $\mathcal{H}$ of interest, $\mathcal{H}$-consistency, for the target 0-1 loss?

**Surrogate losses and Bayes consistency.** For Bayes consistency, the above question is answered by the notion of *calibrated* surrogates [2, 17, 16, 14, 13, 11]. Specifically, if a surrogate loss $\psi$ is calibrated w.r.t. the 0-1 loss, then for any universal function class $\mathcal{F}_{\text{univ}}$, the $(\psi, \mathcal{F}_{\text{univ}})$ surrogate risk minimization algorithm (implemented with suitable regularization) is a Bayes consistent algorithm for $\ell_{\text{0-1}}$.[1] Among the surrogate losses shown in Table 1, $\psi_{\text{mlog}}$ and $\psi_{\text{OvA,log}}$ are universally calibrated for $\ell_{\text{0-1}}$ (calibrated for all probability distributions), while $\psi_{\text{CS}}$ and $\psi_{\text{OvA,hinge}}$ are calibrated under the so-called 'dominant-label' condition (calibrated for distributions in which the conditional distributions $p(y|\mathbf{x})$ assign probability at least $\frac{1}{2}$ to one of the $n$ classes) [16].

**Surrogate losses and $\mathcal{H}$-consistency.** For $\mathcal{H}$-consistency, the situation is more complex [3, 7]. In particular, Long and Servedio [7] showed the following results:[2]

(1) *Realizable $\mathcal{H}_{\text{cls}}$-consistency of Crammer-Singer surrogate for closed-under-scaling models $\mathcal{H}_{\text{cls}}$.* Let $\mathcal{F}_{\text{cls}} \subset \{\mathbf{f} : \mathcal{X} \to \mathbb{R}^n\}$ be any class of (vector) scoring functions that is closed under scaling, and

$$\mathcal{H}_{\text{cls}} = \left\{ h : \mathcal{X} \to \mathcal{Y} \mid \exists \mathbf{f} \in \mathcal{F}_{\text{cls}} \text{ s.t. } h(\mathbf{x}) \in \text{argmax}_{y \in [n]} f_y(\mathbf{x}) \; \forall \mathbf{x} \right\}.$$

Long and Servedio [7] showed that if the data distribution $D$ is $\mathcal{H}_{\text{cls}}$-realizable (i.e. the data is labeled according to a true model $h^* \in \mathcal{H}_{\text{cls}}$), then minimizing the Crammer-Singer surrogate $\psi_{\text{CS}}$ over $\mathcal{F}_{\text{cls}}$ is $\mathcal{H}_{\text{cls}}$-consistent, i.e. the $(\psi_{\text{CS}}, \mathcal{F}_{\text{cls}})$ surrogate risk minimization algorithm is $\mathcal{H}_{\text{cls}}$-consistent. This was viewed as surprising in light of the fact that $\psi_{\text{CS}}$ is not (universally) calibrated for $\ell_{\text{0-1}}$.

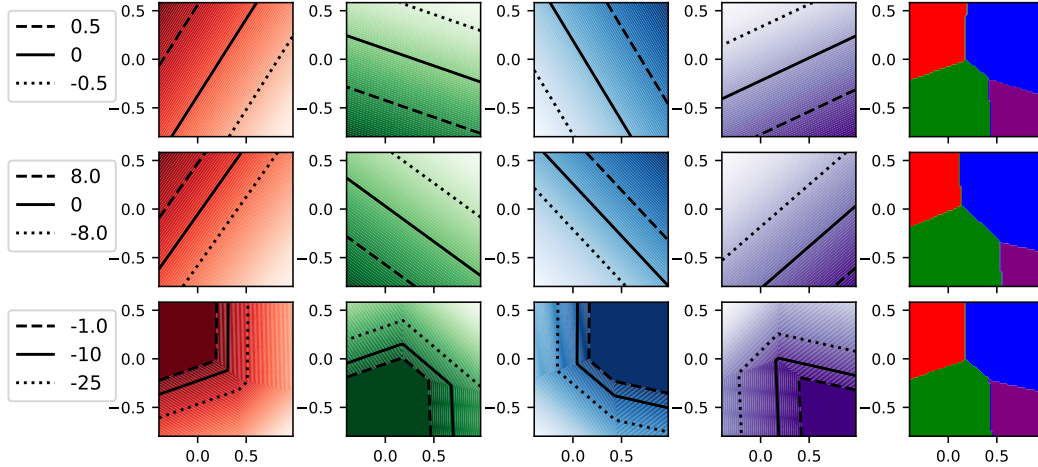

Figure 1: Example in $d = 2$ dimensions with $n = 4$ classes. **Top row:** True linear 4-class classifier $h^* \in \mathcal{H}_{\text{lin}}$. The first 4 plots show contours of the 4 linear scoring functions $f_1^*, \ldots, f_4^* : \mathcal{X} \to \mathbb{R}$ (darker shades represent higher values); the 5th plot shows the regions corresponding to the classifier $h^*(\mathbf{x}) \in \text{argmax}_{y \in [n]} f_y^*(\mathbf{x})$. As can be seen, each class is described by a convex polytope, and is separated from the rest by a piecewise linear decision boundary. **Middle row:** Contours of scoring functions and decision regions learned from training data labeled according to $h^*$ by minimizing the one-vs-all logistic surrogate $\psi_{\text{OvA,log}}$ over the linear scoring function class $\mathcal{F}_{\text{lin}}$. The generalization accuracy is 0.886. **Bottom row:** Contours of scoring functions and decision regions learned from the same training data by minimizing the same one-vs-all logistic surrogate $\psi_{\text{OvA,log}}$ over the 'shared' piecewise linear scoring function class $\mathcal{F}_{\text{spwlin}}$. The decision regions are closer to the true model, and the generalization accuracy is 0.986. Since the piecewise linear functions have shared pieces, the number of parameters to be learned is the same as in the linear case; moreover, the resulting model can also be transformed to a linear model if desired (see Theorem 3 and Corollary 4). See Section 5 for details of the experimental setup.

(2) *Lack of realizable $\mathcal{H}_{\text{lin}}$-consistency of one-vs-all logistic surrogate for linear models $\mathcal{H}_{\text{lin}}$.* Let $\mathcal{F}_{\text{lin}}$ be the class of linear (vector) scoring functions and $\mathcal{H}_{\text{lin}}$ the class of linear multiclass classification models:

$$\mathcal{F}_{\text{lin}} = \left\{ \mathbf{f} : \mathcal{X} \to \mathbb{R}^n \mid \exists \mathbf{w}_1, \ldots, \mathbf{w}_n \in \mathbb{R}^d \text{ s.t. } f_y(\mathbf{x}) = \mathbf{w}_y^\top \mathbf{x} \; \forall \mathbf{x} \right\} \tag{6}$$

$$\mathcal{H}_{\text{lin}} = \left\{ h : \mathcal{X} \to \mathcal{Y} \mid \exists \mathbf{w}_1, \ldots, \mathbf{w}_n \in \mathbb{R}^d \text{ s.t. } h(\mathbf{x}) \in \text{argmax}_{y \in [n]} \mathbf{w}_y^\top \mathbf{x} \; \forall \mathbf{x} \right\}. \tag{7}$$

Long and Servedio [7] showed that even if the data distribution $D$ is $\mathcal{H}_{\text{lin}}$-realizable (i.e. the data is labeled according to a true linear model $h^* \in \mathcal{H}_{\text{lin}}$), minimizing the one-vs-all logistic surrogate $\psi_{\text{OvA,log}}$ over $\mathcal{F}_{\text{lin}}$ fails to give an $\mathcal{H}_{\text{lin}}$-consistent algorithm, i.e. the $(\psi_{\text{OvA,log}}, \mathcal{F}_{\text{lin}})$ surrogate risk minimization algorithm is *not* $\mathcal{H}_{\text{lin}}$-consistent, even though $\psi_{\text{OvA,log}}$ is universally calibrated for $\ell_{\text{0-1}}$.

**Our contributions.** As discussed by Long and Servedio [7] and summarized above, it seems peculiar that the Crammer-Singer surrogate $\psi_{\text{CS}}$, which is not universally calibrated for $\ell_{\text{0-1}}$, provides realizable $\mathcal{H}_{\text{lin}}$-consistency (and more generally, realizable $\mathcal{H}_{\text{cls}}$-consistency for closed-under-scaling models $\mathcal{H}_{\text{cls}}$), while the one-vs-all logistic surrogate, $\psi_{\text{OvA,log}}$, which is universally calibrated for $\ell_{\text{0-1}}$, fails to provide realizable $\mathcal{H}_{\text{lin}}$-consistency. In this paper, we investigate this apparent conundrum.

First, regarding result (1) of Long and Servedio [7] above, we note that any realizable distribution $D$ (i.e. a distribution that labels data points $\mathbf{x}$ according to a deterministic model $y = h(\mathbf{x})$) trivially satisfies the dominant-label condition (for each $\mathbf{x}$, one class $y$ has conditional probability $p(y|\mathbf{x}) \geq \frac{1}{2}$), and therefore the Crammer-Singer surrogate $\psi_{\text{CS}}$ is in fact calibrated for any such distribution. Therefore, in the realizable setting studied by Long and Servedio [7], the surrogate $\psi_{\text{CS}}$ is in fact calibrated for $\ell_{\text{0-1}}$ (the paper emphasizes that $\psi_{\text{CS}}$ is not calibrated/consistent for $\ell_{\text{0-1}}$, implicitly referring to universal calibration, and misses the fact that it is indeed calibrated for the setting studied). So, while the result (1) is still interesting and non-trivial, it should be kept in mind that under the realizable setting studied in [7], all the surrogates studied by the authors are in fact calibrated for $\ell_{\text{0-1}}$.

Second, and more importantly, we look into result (2) of Long and Servedio [7] above. We know that minimizing the one-vs-all logistic surrogate $\psi_{\text{OvA,log}}$ over a universal scoring function class $\mathcal{F}_{\text{univ}}$ gives Bayes consistency for all distributions $D$. Therefore, for $\mathcal{H}_{\text{lin}}$-realizable distributions $D$, where $\text{er}_D^{\text{0-1},*} = \text{er}_D^{\text{0-1}}[\mathcal{H}_{\text{lin}}]$ and therefore Bayes consistency is equivalent to $\mathcal{H}_{\text{lin}}$-consistency, we have that minimizing the $\psi_{\text{OvA,log}}$ surrogate over such a class $\mathcal{F}_{\text{univ}}$ gives an $\mathcal{H}_{\text{lin}}$-consistent algorithm. So why does minimizing the same surrogate over the class $\mathcal{F}_{\text{lin}}$ of linear scoring functions fail in this regard?

On closer inspection, we find that an important part of the answer lies in the form of the decision boundaries induced by a linear (or more generally, affine) multiclass classification model. As an example, Figure 1 shows an affine 4-class model in a 2-dimensional instance space; specifically, the figure shows the 4 affine scoring functions for the 4 classes, and the corresponding decision regions. As can be seen, the one-vs-all boundaries induced by such a model are *not* linear! Indeed, in general, each class is described by a convex polytope, and is separated from the rest of the classes by a piecewise linear decision boundary (where the boundaries for different classes include shared pieces). Therefore, when the one-vs-all classifier is forced to separate each class from the rest using a linear decision boundary, it can end up learning a suboptimal separator.

In the rest of the paper, we use the above insight to design a special class of 'shared' piecewise linear scoring functions $\mathcal{F}_{\mathrm{spwlin}}$ such that minimizing the one-vs-all logistic surrogate $\psi_{\mathrm{OvA,log}}$ over this class yields an $\mathcal{H}_{\mathrm{lin}}$-consistent algorithm. We will see that $\mathcal{F}_{\mathrm{spwlin}}$ is characterized by the same number of parameters as $\mathcal{F}_{\mathrm{lin}}$; in fact, $\mathcal{F}_{\mathrm{spwlin}}$ will also be parametrized by $n$ weight vectors $\mathbf{w}_1, \ldots, \mathbf{w}_n \in \mathbb{R}^d$.[3] In order to minimize $\psi_{\mathrm{OvA,log}}$ over this scoring function class, we will make use of an adaptation of the min-pooling idea from neural network training. The same idea can be applied to other one-vs-all surrogates as well; in our experiments, we consider both $\psi_{\mathrm{OvA,log}}$ and the one-vs-all SVM surrogate $\psi_{\mathrm{OvA,hinge}}$, and find that in both cases, while minimizing these surrogates over the class of linear scoring functions $\mathcal{F}_{\mathrm{lin}}$ fails to provide $\mathcal{H}_{\mathrm{lin}}$-consistency, minimizing them over the *nonlinear* scoring function class $\mathcal{F}_{\mathrm{spwlin}}$ does indeed provide $\mathcal{H}_{\mathrm{lin}}$-consistency.

An additional interesting aspect of the scoring function class $\mathcal{F}_{\mathrm{spwlin}}$ is that, while the individual scoring functions in it are nonlinear (specifically, piecewise linear), the classification models resulting from taking the highest-scoring class according to these scoring functions can also be expressed as linear models. Therefore, having learned a classifier by minimizing a one-vs-all surrogate over this nonlinear scoring function class, one can then convert the learned model to a linear model in $\mathcal{H}_{\mathrm{lin}}$.

We believe our study can pave the way for a more thorough understanding of the role of surrogate losses in $\mathcal{H}$-consistency. In particular, our results suggest that, when studying $\mathcal{H}$-consistency, one needs to carefully take into account the interplay between surrogate losses and the scoring function class over which they are minimized, and that this can lead to unexpected improvements to learning algorithms used in practice.

**Organization.** We start by giving various formal definitions in Section 2. We then describe the class of 'shared' piecewise linear scoring functions $\mathcal{F}_{\mathrm{spwlin}}$ and give our associated consistency result in Section 3. We discuss how to minimize one-vs-all surrogates over this scoring function class in practice in Section 4, and describe our numerical experiments in Section 5. Section 6 concludes with a brief summary. Additional details/proofs are provided in the supplementary material.

## 2 Formal Definitions: Consistency, Calibration, Realizability

**Consistency.** We start with formal definitions of Bayes consistency and $\mathcal{H}$-consistency:

**Definition 1** (Bayes consistency). *We say a multiclass learning algorithm that maps training samples $S \in \cup_{m=1}^{\infty} (\mathcal{X} \times \mathcal{Y})^m$ to multiclass models $\widehat{h} : \mathcal{X} \rightarrow \mathcal{Y}$ is* Bayes consistent *(w.r.t. $\ell_{\text{0-1}}$) for a distribution $D$ if for all $\epsilon > 0$,*

$$\lim_{m \to \infty} \mathbf{P}_{S \sim D^m} \left( \mathrm{er}_D^{\text{0-1}}[\widehat{h}] - \mathrm{er}_D^{\text{0-1},*} > \epsilon \right) \;=\; 0 \,.$$

*If an algorithm is Bayes consistent for* all *distributions $D$, we say it is* universally Bayes consistent.

**Definition 2** ($\mathcal{H}$-consistency). *Let $\mathcal{H} \subset \{h : \mathcal{X} \rightarrow \mathcal{Y}\}$. We say a multiclass learning algorithm that maps training samples $S \in \cup_{m=1}^{\infty} (\mathcal{X} \times \mathcal{Y})^m$ to multiclass models $\widehat{h} : \mathcal{X} \rightarrow \mathcal{Y}$ is* $\mathcal{H}$-consistent *(w.r.t. $\ell_{\text{0-1}}$) for a distribution $D$ if for all $\epsilon > 0$,*

$$\lim_{m \to \infty} \mathbf{P}_{S \sim D^m} \left( \mathrm{er}_D^{\text{0-1}}[\widehat{h}] - \mathrm{er}_D^{\text{0-1}}[\mathcal{H}] > \epsilon \right) \;=\; 0 \,.$$

*Note that we do not require the algorithm to produce a model in $\mathcal{H}$; we only require that as $m \to \infty$, the performance of the model it learns approaches that of the best model in $\mathcal{H}$. If an algorithm is $\mathcal{H}$-consistent for* all *distributions $D$, we say it is* universally $\mathcal{H}$-consistent.

**Calibration.** Next, we give the standard definition of calibration of a surrogate loss that is useful for studying Bayes consistency of surrogate risk minimization algorithms, followed by a definition

of calibration w.r.t. $\mathcal{H}$ that is useful for studying $\mathcal{H}$-consistency of such algorithms. To give these definitions, for a surrogate loss $\psi : \mathcal{Y} \times \mathbb{R}^n \to \mathbb{R}_+$, we need the following notions of $\psi$-*generalization error*, *Bayes optimal $\psi$-error*, and *optimal $\psi$-error in a scoring function class* $\mathcal{F} \subset \{\mathbf{f} : \mathcal{X} \to \mathbb{R}^n\}$:

$$\mathrm{er}_D^\psi[\mathbf{f}] = \mathbf{E}_{(X,Y) \sim D}\big[\psi(Y, \mathbf{f}(X))\big] ; \qquad \mathrm{er}_D^{\psi,*} = \inf_{\mathbf{f}:\mathcal{X}\to\mathbb{R}^n} \mathrm{er}_D^\psi[\mathbf{f}] ; \qquad \mathrm{er}_D^\psi[\mathcal{F}] = \inf_{\mathbf{f}\in\mathcal{F}} \mathrm{er}_D^\psi[\mathbf{f}] . \qquad (8)$$

**Definition 3** (Calibration (standard definition))**.** *We say a surrogate loss* $\psi : \mathcal{Y} \times \mathbb{R}^n \to \mathbb{R}_+$ *is calibrated w.r.t. $\ell_{0\text{-}1}$ for a distribution $D$ if there exists a strictly increasing function $g : \mathbb{R}_+ \to \mathbb{R}_+$ that is continuous at 0 with $g(0) = 0$ such that for all $\mathbf{f} : \mathcal{X} \to \mathbb{R}^n$,*

$$\mathrm{er}_D^{0\text{-}1}\big[\underbrace{\mathrm{argmax} \circ \mathbf{f}}_{h}\big] - \mathrm{er}_D^{0\text{-}1,*} \leq g\Big(\mathrm{er}_D^\psi[\mathbf{f}] - \mathrm{er}_D^{\psi,*}\Big),$$

*where $h \equiv \mathrm{argmax} \circ \mathbf{f}$ denotes a classifier that satisfies $h(\mathbf{x}) \in \mathrm{argmax}_{y\in[n]} f_y(\mathbf{x})$. If $\psi$ is calibrated w.r.t. $\ell_{0\text{-}1}$ for* all *distributions $D$, we say $\psi$ is* universally *calibrated w.r.t. $\ell_{0\text{-}1}$.*

**Definition 4** (Calibration w.r.t. $\mathcal{H}$)**.** *For a class of multiclass models $\mathcal{H} \subset \{h : \mathcal{X} \to \mathcal{Y}\}$, a surrogate loss $\psi : \mathcal{Y} \times \mathbb{R}^n \to \mathbb{R}_+$, and a scoring function class $\mathcal{F} \subset \{\mathbf{f} : \mathcal{X} \to \mathbb{R}^n\}$, we say $(\psi, \mathcal{F})$ is* calibrated *w.r.t. $(\ell_{0\text{-}1}, \mathcal{H})$ for a distribution $D$ if there exists a strictly increasing function $g : \mathbb{R}_+ \to \mathbb{R}_+$ that is continuous at 0 with $g(0) = 0$ such that for all $\mathbf{f} \in \mathcal{F}$,*

$$\mathrm{er}_D^{0\text{-}1}\big[\underbrace{\mathrm{argmax} \circ \mathbf{f}}_{h}\big] - \mathrm{er}_D^{0\text{-}1}[\mathcal{H}] \leq g\Big(\mathrm{er}_D^\psi[\mathbf{f}] - \mathrm{er}_D^\psi[\mathcal{F}]\Big),$$

*where $h \equiv \mathrm{argmax} \circ \mathbf{f}$ denotes a classifier that satisfies $h(\mathbf{x}) \in \mathrm{argmax}_{y\in[n]} f_y(\mathbf{x})$. If $(\psi, \mathcal{F})$ is calibrated w.r.t. $(\ell_{0\text{-}1}, \mathcal{H})$ for* all *distributions $D$, we say $(\psi, \mathcal{F})$ is* universally *calibrated w.r.t. $(\ell_{0\text{-}1}, \mathcal{H})$.*

**Realizability and realizable calibration/consistency.** Finally, we give formal definitions of realizable and $\mathcal{H}$-realizable distributions, realizable calibration, and Long and Servedio's definition of realizable $\mathcal{H}_\mathcal{F}$-consistency.

**Definition 5** (Realizability and $\mathcal{H}$-realizability)**.** *We say a distribution $D$ over $\mathcal{X} \times \mathcal{Y}$ is* realizable *if (almost surely) it labels points according to a deterministic model, i.e. if $\exists h : \mathcal{X} \to \mathcal{Y}$ such that $P_{(X,Y)\sim D}\big(h(X) = Y\big) = 1$. For a class $\mathcal{H} \subset \{h : \mathcal{X} \to \mathcal{Y}\}$, we say a distribution $D$ over $\mathcal{X} \times \mathcal{Y}$ is $\mathcal{H}$-realizable if (almost surely) it labels points according to a deterministic model in $\mathcal{H}$, i.e. if $\exists h \in \mathcal{H}$ such that $P_{(X,Y)\sim D}\big(h(X) = Y\big) = 1$.*

**Definition 6** (Realizable calibration)**.** *We say a surrogate loss $\psi : \mathcal{Y} \times \mathbb{R}^n \to \mathbb{R}_+$ is* realizable calibrated *(w.r.t. $\ell_{0\text{-}1}$) if it is calibrated (w.r.t. $\ell_{0\text{-}1}$) for all realizable distributions.*[4]

**Definition 7** (Long and Servedio's definition of realizable $\mathcal{H}_\mathcal{F}$-consistency [7])**.** *Let $\mathcal{F} \subset \{\mathbf{f} : \mathcal{X} \to \mathbb{R}^n\}$, and let $\mathcal{H}_\mathcal{F} = \{h : \mathcal{X} \to \mathcal{Y} \mid \exists \mathbf{f} \in \mathcal{F} \text{ s.t. } h(\mathbf{x}) \in \mathrm{argmax}_y f_y(\mathbf{x}) \; \forall \mathbf{x}\}$. A surrogate loss $\psi : \mathcal{Y} \times \mathbb{R}^n \to \mathbb{R}_+$ is* realizable $\mathcal{H}_\mathcal{F}$-consistent *if $(\psi, \mathcal{F})$ is calibrated w.r.t. $(\ell_{0\text{-}1}, \mathcal{H}_\mathcal{F})$ for all $\mathcal{H}_\mathcal{F}$-realizable distributions.*[5],[6]

# 3 Minimizing One-vs-All Surrogates over a Class $\mathcal{F}_{\mathbf{spwlin}}$ of 'Shared' Piecewise Linear Scoring Functions is $\mathcal{H}_{\mathbf{lin}}$-Consistent

As discussed in Section 1, even though the one-vs-all logistic surrogate $\psi_{\mathrm{OvA,log}}$ is universally calibrated for $\ell_{0\text{-}1}$, Long and Servedio [7] showed that the $(\psi_{\mathrm{OvA,log}}, \mathcal{F}_{\mathrm{lin}})$ surrogate risk minimization algorithm, which minimizes $\psi_{\mathrm{OvA,log}}$ over the class of linear scoring functions $\mathcal{F}_{\mathrm{lin}}$, is not $\mathcal{H}_{\mathrm{lin}}$-consistent even when the data distribution $D$ is $\mathcal{H}_{\mathrm{lin}}$-realizable. In this section, we remedy this situation by showing how to minimize the same surrogate loss $\psi_{\mathrm{OvA,log}}$ (as well as other one-vs-all surrogate losses) over a different, nonlinear scoring function class $\mathcal{F}_{\mathrm{spwlin}}$ such that the resulting algorithm is $\mathcal{H}_{\mathrm{lin}}$-consistent for all $\mathcal{H}_{\mathrm{lin}}$-realizable distributions $D$.

**Linear models.** For the remainder of the paper, we will re-define the classes of linear scoring functions and linear classification models to allow for the inclusion of bias/offset terms:

$$\mathcal{F}_{\text{lin}} = \left\{ \mathbf{f} : \mathcal{X} \to \mathbb{R}^n \mid \exists \mathbf{w}_1, \ldots, \mathbf{w}_n \in \mathbb{R}^d, b_1 \ldots, b_n \in \mathbb{R} \text{ s.t. } f_y(\mathbf{x}) = \mathbf{w}_y^\top \mathbf{x} + b_y \ \forall \mathbf{x} \right\} \tag{9}$$

$$\mathcal{H}_{\text{lin}} = \left\{ h : \mathcal{X} \to \mathcal{Y} \mid \exists \mathbf{w}_1, \ldots, \mathbf{w}_n \in \mathbb{R}^d, b_1 \ldots, b_n \in \mathbb{R} \text{ s.t. } h(\mathbf{x}) \in \operatorname{argmax}_{y \in [n]} \mathbf{w}_y^\top \mathbf{x} + b_y \ \forall \mathbf{x} \right\}. \tag{10}$$

Our conclusions will apply both in this more general setting, and in the special case where $b_y = 0 \ \forall y$.

**'Shared' piecewise linear scoring functions.** To motivate the scoring function class we will construct, consider again the example in Figure 1. As this example makes clear, under a linear classification model in $\mathcal{H}_{\text{lin}}$ defined by weight vectors $\mathbf{w}_1, \ldots, \mathbf{w}_n \in \mathbb{R}^d$ and bias terms $b_1 \ldots, b_n \in \mathbb{R}$, the decision region corresponding exclusively to class $y \in [n]$ is the (open) convex polytope given by

$$\mathcal{X}_y = \left\{ \mathbf{x} \in \mathcal{X} \mid \mathbf{w}_y^\top \mathbf{x} + b_y > \mathbf{w}_{y'}^\top \mathbf{x} + b_{y'} \ \forall y' \neq y \right\} \tag{11}$$

$$= \left\{ \mathbf{x} \in \mathcal{X} \mid (\mathbf{w}_y - \mathbf{w}_{y'})^\top \mathbf{x} + (b_y - b_{y'}) > 0 \ \forall y' \neq y \right\} \tag{12}$$

$$= \left\{ \mathbf{x} \in \mathcal{X} \mid \min_{y' \neq y} \left\{ (\mathbf{w}_y - \mathbf{w}_{y'})^\top \mathbf{x} + (b_y - b_{y'}) \right\} > 0 \right\}. \tag{13}$$

We use this observation to construct the following special class of 'shared' piecewise linear scoring functions:

$$\mathcal{F}_{\text{spwlin}} = \left\{ \mathbf{f} : \mathcal{X} \to \mathbb{R}^n \mid \exists \mathbf{w}_1, \ldots, \mathbf{w}_n \in \mathbb{R}^d, b_1 \ldots, b_n \in \mathbb{R} \text{ s.t. } \right.$$
$$\left. f_y(\mathbf{x}) = \min_{y' \neq y} \left\{ (\mathbf{w}_y - \mathbf{w}_{y'})^\top \mathbf{x} + (b_y - b_{y'}) \right\} \ \forall \mathbf{x} \right\}. \tag{14}$$

Clearly, this class is parametrized by the same number of parameters as $\mathcal{F}_{\text{lin}}$. The reason that the class $\mathcal{F}_{\text{spwlin}}$ is useful is that the scoring functions in this class will allow for learning precisely the form of one-vs-all decision boundaries that are induced by linear multiclass models. In particular, we have the following result:

**Lemma 1** (Scoring functions in $\mathcal{F}_{\text{spwlin}}$ capture correct one-vs-all decision boundaries for linear multiclass models)**.** *Let* $\mathbf{f} \in \mathcal{F}_{\text{spwlin}}$ *be parametrized by* $\mathbf{w}_1, \ldots, \mathbf{w}_n \in \mathbb{R}^d, b_1, \ldots, b_n \in \mathbb{R}$. *Then*

$$f_y(\mathbf{x}) \geq 0 \iff y \in \operatorname{argmax}_{y' \in [n]} \mathbf{w}_{y'}^\top \mathbf{x} + b_{y'}. \tag{15}$$

Since one-vs-all surrogates effectively learn scoring functions that aim to separate points $\mathbf{x}$ with label $y$ from points with other labels according to whether $f_y(\mathbf{x}) \geq 0$, the above result implies that minimizing such surrogates over the class $\mathcal{F}_{\text{spwlin}}$ should allow learning precisely the form of one-vs-all separation boundaries induced by linear multiclass models. Formally, we have the following $\mathcal{H}_{\text{lin}}$-consistency result:

**Theorem 2** ($\mathcal{H}_{\text{lin}}$-consistency of $(\psi_{\text{OvA,log}}, \mathcal{F}_{\text{spwlin}})$ surrogate risk minimization algorithm)**.** *The pair* $(\psi_{\text{OvA,log}}, \mathcal{F}_{\text{spwlin}})$ *is calibrated w.r.t.* $(\ell_{\text{0-1}}, \mathcal{H}_{\text{lin}})$ *for all* $\mathcal{H}_{\text{lin}}$-*realizable distributions.*

**Remark 1** (Generalization to other one-vs-all surrogates)**.** *The above* $\mathcal{H}_{\text{lin}}$-*consistency result can be generalized to other one-vs-all surrogates, such as the one-vs-all hinge surrogate* $\psi_{\text{OvA,hinge}}$.

**Remark 2** (Loss of 'independence' of one-vs-all binary classifiers)**.** *Since the $n$ components of the (vector) scoring functions in $\mathcal{F}_{\text{spwlin}}$ share parameters, they can no longer be learned independently by training separate binary classifiers in parallel; while minimizing a one-vs-all surrogate over $\mathcal{F}_{\text{spwlin}}$ still amounts to learning binary separators for each of the classes versus the rest, these separators must be learned together in an "all-in-one" multiclass learning algorithm.*

**Remark 3** (Non-convexity of resulting optimization problems)**.** *Although the one-vs-all surrogates $\psi_{\text{OvA,log}}$ and $\psi_{\text{OvA,hinge}}$ are convex, minimizing these surrogates over the function class $\mathcal{F}_{\text{spwlin}}$ results in non-convex optimization problems. In order to solve these optimization problems, our implementation makes use of an adaptation of the min-pooling idea from neural network training (see Section 4). Additional details regarding the behavior of this approach in our experiments are discussed in Section 5.*

We also have the following result, which shows that the classification models induced by the nonlinear (vector) scoring functions in $\mathcal{F}_{\text{spwlin}}$ are in fact equivalent to those in the class of *linear* classification models $\mathcal{H}_{\text{lin}}$:

**Theorem 3** (Scoring functions in $\mathcal{F}_{\text{spwlin}}$ induce linear multiclass classifiers)**.** *Let $\mathcal{H}_{\text{spwlin}}$ be the class of multiclass classifiers induced by $\mathcal{F}_{\text{spwlin}}$:*

$$\mathcal{H}_{\text{spwlin}} = \left\{ h : \mathcal{X} \to \mathcal{Y} \mid \exists \mathbf{f} \in \mathcal{F}_{\text{spwlin}} \text{ s.t. } h(\mathbf{x}) \in \operatorname{argmax}_{y \in [n]} f_y(\mathbf{x}) \right\}. \tag{16}$$

*Then* $\mathcal{H}_{\text{spwlin}} = \mathcal{H}_{\text{lin}}$.

Indeed, the following corollary shows that once we have learned a nonlinear (vector) scoring function in $\mathcal{F}_{\text{spwlin}}$, we can easily transform it into a linear classification model in $\mathcal{H}_{\text{lin}}$:

**Corollary 4** (Converting a nonlinear scoring function in $\mathcal{F}_{\text{spwlin}}$ to a linear classification model in $\mathcal{H}_{\text{lin}}$)**.** *Let* $\mathbf{f} \in \mathcal{F}_{\text{spwlin}}$ *be parametrized by* $\mathbf{w}_1, \ldots, \mathbf{w}_n \in \mathbb{R}^d, b_1, \ldots, b_n \in \mathbb{R}$. *Then for all* $\mathbf{x} \in \mathcal{X}$,
$$\text{argmax}_{y \in [n]}\, f_y(\mathbf{x}) \;=\; \text{argmax}_{y \in [n]}\, \mathbf{w}_y^\top \mathbf{x} + b_y \,. \tag{17}$$

**Margin interpretation of $\mathcal{F}_{\text{spwlin}}$.** We note that the scoring functions in $\mathcal{F}_{\text{spwlin}}$ can also be viewed as computing a multiclass 'margin' vector over the underlying linear functions defining the shared piecewise linear scores. Specifically, recall that a (vector) scoring function $\mathbf{f} \in \mathcal{F}_{\text{spwlin}}$ has the form
$$f_y(\mathbf{x}) \;=\; \min_{y' \neq y}\left\{ (\mathbf{w}_y - \mathbf{w}_{y'})^\top \mathbf{x} + (b_y - b_{y'}) \right\} \;=\; (\mathbf{w}_y^\top \mathbf{x} + b_y) - \max_{y' \neq y}\left\{ \mathbf{w}_{y'}^\top \mathbf{x} + b_{y'} \right\} \tag{18}$$
for some $\{\mathbf{w}_y, b_y\}_{y=1}^n$. This suggests that for each $y$, the score $f_y(\mathbf{x})$ effectively computes the 'margin' of separation between $(\mathbf{w}_y^\top \mathbf{x} + b_y)$ and $\max_{y' \neq y}\{\mathbf{w}_y^\top \mathbf{x} + b_{y'}\}$; if this margin is non-negative, then $y \in \text{argmax}_{y' \in [n]} \mathbf{w}_{y'}^\top \mathbf{x} + b_{y'}$, and if it is negative, then $y \notin \text{argmax}_{y' \in [n]} \mathbf{w}_{y'}^\top \mathbf{x} + b_{y'}$.

**Generalization to other multiclass models $\mathcal{H}$.** The above construction can be generalized beyond $\mathcal{H}_{\text{lin}}$ to other multiclass models $\mathcal{H}_{\mathcal{G}}$ defined in terms of a class of real-valued scoring functions $\mathcal{G} \subset \{g : \mathcal{X} \to \mathbb{R}\}$. Specifically, for any such class $\mathcal{G}$, let
$$\mathcal{H}_{\mathcal{G}} \;=\; \left\{ h : \mathcal{X} \to \mathcal{Y} \mid \exists g_1, \ldots, g_n \in \mathcal{G} \text{ s.t. } h(\mathbf{x}) \in \text{argmax}_{y \in [n]}\, g_y(\mathbf{x})\ \forall \mathbf{x} \right\} \,. \tag{19}$$
(Thus $\mathcal{H}_{\text{lin}}$ is a special case with $\mathcal{G}_{\text{lin}} = \{g : \mathcal{X} \to \mathbb{R} \mid \exists \mathbf{w} \in \mathbb{R}^d, b \in \mathbb{R} \text{ s.t. } g(\mathbf{x}) = \mathbf{w}^\top \mathbf{x} + b\ \forall \mathbf{x}\}$.)
Define the class of 'shared' piecewise-difference-of-$\mathcal{G}$ scoring functions $\mathcal{F}_{\text{spwdiff}\mathcal{G}}$ as follows:
$$\mathcal{F}_{\text{spwdiff}\mathcal{G}} \;=\; \left\{ \mathbf{f} : \mathcal{X} \to \mathbb{R}^n \mid \exists g_1, \ldots, g_n \in \mathcal{G} \text{ s.t. } f_y(\mathbf{x}) = \min_{y' \neq y}\{ g_y(\mathbf{x}) - g_{y'}(\mathbf{x}) \}\ \forall \mathbf{x} \right\} \,. \tag{20}$$
Then similarly to the linear case, it can be shown that minimizing any of the one-vs-all surrogates $\psi_{\text{OvA,log}}$ or $\psi_{\text{OvA,hinge}}$ over $\mathcal{F}_{\text{spwdiff}\mathcal{G}}$ is $\mathcal{H}_{\mathcal{G}}$-consistent for all $\mathcal{H}_{\mathcal{G}}$-realizable distributions.

# 4    Implementation of One-vs-All Surrogate Risk Minimization over $\mathcal{F}_{\text{spwlin}}$

In order to implement surrogate risk minimization over the scoring function class $\mathcal{F}_{\text{spwlin}}$, we make use of an adaptation of the min-pooling idea from neural network training. Figure 2 shows a summary of the architecture we use to implement scoring functions $\mathbf{f}$ in $\mathcal{F}_{\text{spwlin}}$.

Specifically, given an input point $\mathbf{x} \in \mathbb{R}^d$, the first layer computes the $n$ linear functions
$$g_y(\mathbf{x}) = \mathbf{w}_y^\top \mathbf{x} + b_y \,, \quad y \in [n] \,.$$
The second layer then computes the $n$ scoring function components $f_y(\mathbf{x})$ in terms of minima of the relevant functions from the first layer (see Eq. (14)):
$$\mu_y(\mathbf{g}) = \min_{y' \neq y}\left\{ g_y - g_{y'} \right\} \,, \quad y \in [n] \,.$$
To fit the parameters $\{\mathbf{w}_y, b_y\}_{y=1}^n$ to training data, we then use a backpropagation-like procedure to minimize the surrogate loss of interest. Any existing neural network training library can be easily modified to perform this minimization; in our experiments, we implemented this approach using PyTorch [10].

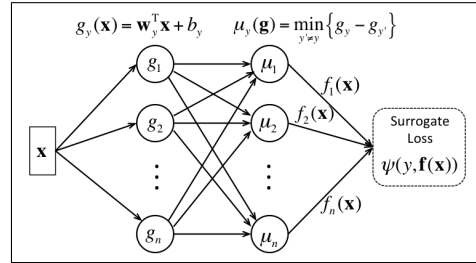

Figure 2: Neural network-like architecture implementing scoring functions in $\mathcal{F}_{\text{spwlin}}$. To find parameters $\{\mathbf{w}_y, b_y\}_{y=1}^n$ minimizing a surrogate loss $\psi$ on the training data, we use a backpropagation-like procedure on this architecture.

# 5    Experiments

We conducted two sets of experiments. In the first set, we generated synthetic data from a true linear model (i.e. a known $\mathcal{H}_{\text{lin}}$-realizable distribution) and tested the $\mathcal{H}_{\text{lin}}$-consistency of minimizing one-vs-all surrogates over $\mathcal{F}_{\text{spwlin}}$. In the second set, we implemented the approach on various real benchmark data sets to test its practical behavior. In all cases, we implemented a total of 6 multiclass algorithms: all 4 algorithms shown in Table 1 with surrogate risk minimized over linear scoring functions $\mathcal{F}_{\text{lin}}$, and the two one-vs-all algorithms with surrogate risk minimized over $\mathcal{F}_{\text{spwlin}}$. All algorithms were implemented in PyTorch and used the AdamW optimizer [8].[7],[8]

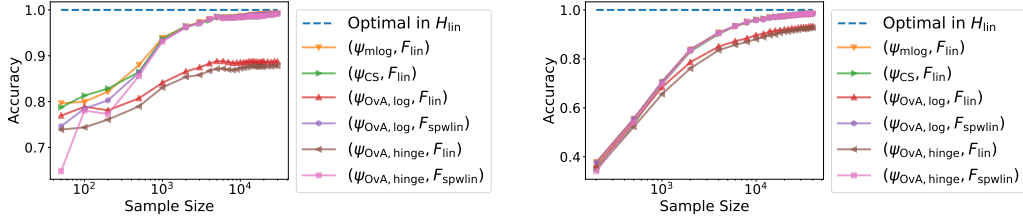

Figure 3: Convergence behavior of various multiclass surrogate risk minimization algorithms on synthetic data generated from a true linear model in $\mathcal{H}_{\text{lin}}$. **Left:** $d = 2$, $n = 4$. **Right:** $d = 100$, $n = 10$. In both cases, the $\psi_{\text{OvA,log}}$ and $\psi_{\text{OvA,hinge}}$ surrogates fail to converge to the optimal performance in $\mathcal{H}_{\text{lin}}$ when minimized over standard linear scoring functions $\mathcal{F}_{\text{lin}}$, but successfully do so when minimized over the class $\mathcal{F}_{\text{spwlin}}$. (In the right plot, the curves for all four $\mathcal{H}_{\text{lin}}$-consistent algorithms overlap.) See Section 5.1 for details.

## 5.1 Synthetic Data: Consistency Behavior on Linear Models

We generated two synthetic data sets. The first data set had $d = 2$ features and $n = 4$ classes. A true model $h^* \in \mathcal{H}_{\text{lin}}$ was created by choosing $\{\mathbf{w}_y, b_y\}_{y=1}^4$ as follows: elements of $\mathbf{w}_y \in \mathbb{R}^2$ were drawn i.i.d. from $\mathcal{N}(0, 1)$ and subsequently scaled so that $\|\mathbf{w}_y\|_2 = 1 \; \forall y$; bias terms $b_1, \ldots, b_4$ were set to $0.2, 0.1, -0.1, -0.2$ (decision regions of the resulting model $h^*$ are shown in Figure 1). Instances $\mathbf{x}$ were then drawn uniformly at random from a disk of radius 0.5 centered at $(0.3, -0.1)$, and labeled according to $h^*$. We ran all 6 algorithms (using AdamW with zero weight decay factor) on increasingly large training samples (up to 30,000 data points) generated in this manner, and measured the generalization accuracy on a large test set of 10,000 data points generated in the same manner. The results are shown in Figure 3 (left); an illustration of some of the models learned from 10,000 data points is also shown in Figure 1.

The second data set had $d = 100$ features and $n = 10$ classes. A true model $h^* \in \mathcal{H}_{\text{lin}}$ was created in the same manner as above, except that in this case we set $b_y = 0 \; \forall y \in [100]$. Instances $\mathbf{x}$ were drawn uniformly at random from $\mathcal{X} = [-1, 1]^{100}$, and labeled according to $h^*$. We ran all 6 algorithms on increasingly large training samples (up to $40,000$ data points) and measured accuracy on a large test set of $10,000$ data points. The results are shown in Figure 3 (right).

In both cases, the one-vs-all surrogates fail to give $\mathcal{H}_{\text{lin}}$-consistency when minimized over linear scoring functions $\mathcal{F}_{\text{lin}}$, but successfully do so when minimized over the scoring function class $\mathcal{F}_{\text{spwlin}}$.

## 5.2 Real Data: Practical Behavior

We evaluated the performance of all 6 algorithms on various benchmark multiclass classification data sets drawn from the UCI repository and the LIBSVM data repository.[9] Details of the data sets are provided in the supplementary material; the number of features $d$ ranges from 16 to 3072, and the number of classes $n$ ranges from 7 to 26. Several of the data sets come with prescribed train/validation/test splits; for the others, we randomly chose a 3:1:1 split. For all algorithms, we used AdamW with a weight decay factor $\lambda$; the factor $\lambda$ was chosen from $\{10^{-3}, ..., 10^2\}$ to maximize 0-1 accuracy on the validation set.

Table 2: Results (in terms of test accuracy) on various real multiclass data sets. See Section 5.2 for details.

| **Data set** | $\psi_{\text{mlog}}$ $\mathcal{F}_{\text{lin}}$ | $\psi_{\text{OvA,log}}$ $\mathcal{F}_{\text{lin}}$ | $\psi_{\text{OvA,log}}$ $\mathcal{F}_{\text{spwlin}}$ | $\psi_{\text{CS}}$ $\mathcal{F}_{\text{lin}}$ | $\psi_{\text{OvA,hinge}}$ $\mathcal{F}_{\text{lin}}$ | $\psi_{\text{OvA,hinge}}$ $\mathcal{F}_{\text{spwlin}}$ |
|---|---|---|---|---|---|---|
| Covertype (50K) | 0.6606 | **0.6943** | 0.6607 | 0.7186 | 0.7069 | *__0.7193__* |
| Digits | **0.8985** | 0.8696 | 0.8982 | 0.9025 | 0.8819 | *__0.9042__* |
| USPS | *__0.9153__* | 0.9138 | 0.9148 | 0.9128 | 0.9063 | **0.9148** |
| MNIST (70K) | 0.9270 | 0.9200 | **0.9271** | 0.9307 | 0.9216 | *__0.9317__* |
| CIFAR10 | 0.4000 | *__0.4066__* | 0.3763 | 0.3831 | 0.3686 | **0.4006** |
| Sensorless | *__0.8266__* | 0.6539 | 0.7918 | 0.7703 | 0.5381 | **0.7791** |
| Letter | 0.7644 | 0.7126 | **0.7662** | 0.7738 | 0.6058 | *__0.7804__* |

The results are shown in Table 2. For each data set, the best-performing algorithms within the group of logistic surrogates and within the group of hinge surrogates are shown in bold font; the best overall is enclosed in asterisks. For hinge surrogates, consistent with previous results [5], we find $\psi_{\text{OvA,hinge}}$,

when minimized over $\mathcal{F}_{\mathrm{lin}}$, does slightly poorer than $\psi_{\mathrm{CS}}$, but minimizing it over $\mathcal{F}_{\mathrm{spwlin}}$ brings it in line with (and even slightly exceeds) $\psi_{\mathrm{CS}}$. For logistic surrogates, the results are more mixed, although $(\psi_{\mathrm{OvA,log}}, \mathcal{F}_{\mathrm{spwlin}})$ frequently outperforms $(\psi_{\mathrm{OvA,log}}, \mathcal{F}_{\mathrm{lin}})$. Overall, despite the good performance, we do not necessarily advocate minimizing one-vs-all surrogates over $\mathcal{F}_{\mathrm{spwlin}}$ as a practical strategy, as training is 2-3 times slower than for $\psi_{\mathrm{CS}}$ or $\psi_{\mathrm{mlog}}$, which generally give comparable results. Our primary interest is in the $\mathcal{H}_{\mathrm{lin}}$-consistency of this scheme under $\mathcal{H}_{\mathrm{lin}}$-realizable data distributions; the main purpose of the experiments on real data was to serve as a sanity check and ensure that this does not come at a huge price in terms of practical applicability of the resulting algorithms.

## 6   Conclusion

Our study shows that when studying $\mathcal{H}$-consistency of surrogate risk minimization algorithms, the interplay between the surrogate loss and scoring function class can play an important role. In particular, for $\psi_{\mathrm{OvA,log}}$ and $\psi_{\mathrm{OvA,hinge}}$, we found that minimization over a suitable function class $\mathcal{F}_{\mathrm{spwlin}}$ gives $\mathcal{H}_{\mathrm{lin}}$-consistency where standard minimization over linear functions $\mathcal{F}_{\mathrm{lin}}$ fails to do so.

### Broader Impact

The primary goal of this paper is to better understand the statistical consistency properties of surrogate risk minimization algorithms in machine learning. The insights and results of the paper will benefit readers who wish to be aware of these properties when designing or selecting learning algorithms.

We do not expect this research to put anyone at a disadvantage. Nevertheless, issues related to data bias and fairness can potentially affect any algorithm that learns models from data [9], and users should keep this in mind when applying the ideas discussed here to domains where such issues may be important. In the future, it may also be of interest to consider incorporating fairness constraints in the types of algorithms discussed here.

### Acknowledgments and Disclosure of Funding

Thanks to Avrim Blum for early discussions related to this work. Part of the motivation for this work also came from discussions following a talk by SA at a workshop on machine learning theory held at Google NYC in September 2019; thanks to all the participants of the workshop for stimulating discussions. We also thank the anonymous referees for helpful comments.

This material is based upon work supported in part by the US National Science Foundation (NSF) under Grant No. 1934876. SA is also supported in part by the US National Institutes of Health (NIH) under Grant No. U01CA214411. Any opinions, findings, and conclusions or recommendations expressed in this material are those of the authors and do not necessarily reflect the views of the NSF or NIH.

## Footnotes

[1] A universal function class is one that can approximate any continuous function; such classes can be obtained, for example, via reproducing kernel Hilbert spaces (RKHSs) associated with Gaussian kernels [12], or via sufficiently flexible neural networks [1].

[2] Long and Servedio [7] presented the results slightly differently; in particular, in their case, $\mathcal{H}$ refers to a class of real-valued functions from which individual scoring functions are drawn, and consistency is defined in terms of this class. We describe the results here in terms of our notation and terminology.

[3]More generally, we will allow both the linear and piecewise linear classes to be characterized by $n$ weight vectors $\mathbf{w}_1, \ldots, \mathbf{w}_n \in \mathbb{R}^d$ and $n$ bias/offset terms $b_1, \ldots, b_n \in \mathbb{R}$.

[4]This is the sense used in Table 1, column 4.

[5]Technically, Long and Servedio's definition [7] applies to scoring function classes $\mathcal{F}$ for which individual scoring function components come independently from a common fixed class, i.e. for which there is a class $\mathcal{F}_0 \subset \{f : \mathcal{X} \to \mathbb{R}\}$ such that $\mathcal{F} = \{\mathbf{f} : \mathcal{X} \to \mathbb{R}^n \mid f_y \in \mathcal{F}_0 \; \forall y\}$, and they would refer to such a surrogate as *realizable $\mathcal{F}_0$-consistent*. We modify the terminology slightly to better fit our presentation of ideas, and the definition we give is slightly more general (in that it allows for more general scoring function classes $\mathcal{F}$).

[6]This is the sense used in Table 1, column 5.

[7]As noted above, the minimization over $\mathcal{F}_{\text{spwlin}}$ is non-convex; we found that for most (but not all) data sets, the results were fairly stable under different random initializations. The results we report are for a single random initialization; our results could potentially be improved by starting the optimizer from multiple random initializations, and keeping the model with best training objective value.

[8]In all cases, the optimizer was run for 50 epochs over the training sample; the learning rate parameter $\alpha$ was initially set to 0.01 and was halved at the end of every 5 epochs.

[9]https://archive.ics.uci.edu/ml/index.php   and   https://www.csie.ntu.edu.tw/~cjlin/libsvmtools/datasets/

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
