[Supplementary Material]

# Bayes Consistency vs. $\mathcal{H}$-Consistency:
# The Interplay between Surrogate Loss Functions and
# the Scoring Function Class

## Appendix

## A  Proof of Lemma 1

*Proof.* This essentially follows from the definition of $\mathcal{F}_{\text{spwlin}}$. In particular, we have:

$$
\begin{aligned}
f_y(\mathbf{x}) \geq 0 \quad &\Longleftrightarrow \quad \min_{y' \neq y} \left\{ (\mathbf{w}_y - \mathbf{w}_{y'})^\top \mathbf{x} + (b_y - b_{y'}) \right\} \geq 0 \\
&\Longleftrightarrow \quad \min_{y' \neq y} \left\{ (\mathbf{w}_y^\top \mathbf{x} + b_y) - (\mathbf{w}_{y'}^\top \mathbf{x} + b_{y'}) \right\} \geq 0 \\
&\Longleftrightarrow \quad (\mathbf{w}_y^\top \mathbf{x} + b_y) \geq (\mathbf{w}_{y'}^\top \mathbf{x} + b_{y'}) \quad \forall y' \neq y \\
&\Longleftrightarrow \quad y \in \text{argmax}_{y' \in [n]} \, \mathbf{w}_{y'}^\top \mathbf{x} + b_{y'} \,.
\end{aligned}
$$

$\square$

## B  Proof of Theorem 2

*Proof.* Let $D$ be a $\mathcal{H}_{\text{lin}}$-realizable distribution. Then $\exists h^* \in \mathcal{H}_{\text{lin}}$ such that $\mathbf{P}_{(X,Y) \sim D}(Y = h^*(X)) = 1$, and therefore $\text{er}_D^{\text{0-1}}[\mathcal{H}_{\text{lin}}] = 0$. Thus our goal is to show that $\exists$ a strictly increasing function $g : \mathbb{R}_+ \to \mathbb{R}_+$ that is continuous at 0 with $g(0) = 0$ such that for all $\mathbf{f} \in \mathcal{F}_{\text{spwlin}}$,

$$
\text{er}_D^{\text{0-1}}[\text{argmax} \circ \mathbf{f}] \leq g\left( \text{er}_D^{\text{OvA,log}}[\mathbf{f}] - \text{er}_D^{\text{OvA,log}}[\mathcal{F}_{\text{spwlin}}] \right).
$$

We will do this in two parts:

(1) We will show that $\text{er}_D^{\text{OvA,log}}[\mathcal{F}_{\text{spwlin}}] = 0$.

(2) We will show that for all $\mathbf{f} \in \mathcal{F}_{\text{spwlin}}$,  $\text{er}_D^{\text{0-1}}[\text{argmax} \circ \mathbf{f}] \leq \frac{1}{\ln(2)} \text{er}_D^{\text{OvA,log}}[\mathbf{f}]$.

Putting these together will then give that for all $\mathbf{f} \in \mathcal{F}_{\text{spwlin}}$,

$$
\text{er}_D^{\text{0-1}}[\text{argmax} \circ \mathbf{f}] \leq \frac{1}{\ln(2)} \left( \text{er}_D^{\text{OvA,log}}[\mathbf{f}] - \text{er}_D^{\text{OvA,log}}[\mathcal{F}_{\text{spwlin}}] \right).
$$

**Part 1.** We will show that for any sufficiently small $\epsilon > 0$, $\exists \mathbf{f}^\epsilon \in \mathcal{F}_{\text{spwlin}}$ such that $\text{er}_D^{\text{OvA,log}}[\mathbf{f}^\epsilon] < \epsilon$; this will establish that $\text{er}_D^{\text{OvA,log}}[\mathcal{F}_{\text{spwlin}}] = 0$.

Let $0 < \epsilon < 2n \ln(2)$. Since $h^* \in \mathcal{H}_{\text{lin}}$, we have $\exists \{\mathbf{w}_y^*, b_y^*\}_{y=1}^n$ such that

$$
h^*(\mathbf{x}) \in \text{argmax}_{y \in [n]} \, (\mathbf{w}_y^*)^\top \mathbf{x} + b_y^* \quad \forall \mathbf{x} \,.
$$

Define $\mathbf{f}^* \in \mathcal{F}_{\text{spwlin}}$ as

$$
\begin{aligned}
f_y^*(\mathbf{x}) &= \min_{y' \neq y} \left\{ (\mathbf{w}_y^* - \mathbf{w}_{y'}^*)^\top \mathbf{x} + (b_y^* - b_{y'}^*) \right\} \\
&= \min_{y' \neq y} \left\{ ((\mathbf{w}_y^*)^\top \mathbf{x} + b_y^*) - ((\mathbf{w}_{y'}^*)^\top \mathbf{x} + b_{y'}^*) \right\} \,.
\end{aligned}
$$

Then we have

$$
\mathbf{P}_{(X,Y) \sim D}\left( f_Y^*(X) > 0 \right) = 1 \,.
$$

Therefore $\exists \kappa > 0$ such that

$$
\mathbf{P}_{(X,Y) \sim D}\left( f_Y^*(X) < \kappa \right) \leq \frac{\epsilon}{2n \ln(2)} \,.
$$

Define $\mathbf{f}^\epsilon \in \mathcal{F}_{\text{spwlin}}$ as

$$f_y^\epsilon(\mathbf{x}) \;=\; \frac{f_y^*(\mathbf{x})}{\kappa}\ln\left(\frac{1}{e^{\epsilon/2n}-1}\right).$$

Then it can be verified that

$$f_y^*(\mathbf{x}) > 0 \;\implies\; f_y^\epsilon(\mathbf{x}) > 0 \;\implies\; \psi_{\text{OvA,log}}(y,\mathbf{f}^\epsilon(\mathbf{x})) \le n\ln(2)\,,$$

and moreover,

$$f_y^*(\mathbf{x}) \ge \kappa \;\implies\; f_y^\epsilon(\mathbf{x}) \ge \ln\left(\frac{1}{e^{\epsilon/2n}-1}\right) \;\implies\; \psi_{\text{OvA,log}}(y,\mathbf{f}^\epsilon(\mathbf{x})) \le \frac{\epsilon}{2}\,.$$

This gives

$$
\begin{aligned}
\text{er}_D^{\text{OvA,log}}[\mathbf{f}^\epsilon] &= \mathbf{E}_{(X,Y)\sim D}\big[\psi_{\text{OvA,log}}\big(Y,\mathbf{f}^\epsilon(X)\big)\big]\\
&\le \mathbf{P}_{(X,Y)\sim D}\big(0 < f_Y^*(X) < \kappa\big)\cdot \mathbf{E}\big[\psi_{\text{OvA,log}}\big(Y,\mathbf{f}^\epsilon(X)\big)\,\big|\,0 < f_Y^*(X) < \kappa\big]\\
&\quad + \mathbf{P}_{(X,Y)\sim D}\big(f_Y^*(X) \ge \kappa\big)\cdot \mathbf{E}\big[\psi_{\text{OvA,log}}\big(Y,\mathbf{f}^\epsilon(X)\big)\,\big|\,f_Y^*(X) \ge \kappa\big]\\
&\le \frac{\epsilon}{2n\ln(2)}\cdot n\ln(2) + 1\cdot\frac{\epsilon}{2}\\
&= \epsilon\,.
\end{aligned}
$$

**Part 2.** Let $\mathbf{f} \in \mathcal{F}_{\text{spwlin}}$, and let $\{\mathbf{w}_y,b_y\}_{y=1}^n$ be such that

$$f_y(\mathbf{x}) \;=\; \min_{y'\ne y}\big\{(\mathbf{w}_y - \mathbf{w}_{y'})^\top\mathbf{x} + (b_y - b_{y'})\big\}\quad \forall \mathbf{x}\,.$$

Define $h : \mathcal{X}\to\mathcal{Y}$ such that

$$h(\mathbf{x}) \;\in\; \text{argmax}_{y\in[n]}f_y(\mathbf{x})\quad \forall\mathbf{x}\,.$$

Then we have

$$
\begin{aligned}
\text{er}_D^{\text{0-1}}[h] &= \mathbf{E}_{(X,Y)\sim D}\big[\ell_{\text{0-1}}(Y,h(X))\big]\\
&= \mathbf{E}_{(X,Y)\sim D}\big[\mathbf{1}\big(h(X)\ne Y\big)\big]\\
&= \mathbf{E}_{(X,Y)\sim D}\left[\sum_{y\ne Y}\mathbf{1}\big(h(X)=y\big)\right]\\
&\le \mathbf{E}_{(X,Y)\sim D}\left[\sum_{y\ne Y}\mathbf{1}\big(f_y(X)\ge 0\big)\right]\quad \text{(by definition of } h \text{ and Lemma 1)}\\
&\le \frac{1}{\ln(2)}\,\mathbf{E}_{(X,Y)\sim D}\left[\sum_{y\ne Y}\ln\big(1+e^{f_y(X)}\big)\right]\\
&\le \frac{1}{\ln(2)}\,\mathbf{E}_{(X,Y)\sim D}\left[\ln\big(1+e^{-f_Y(X)}\big) + \sum_{y\ne Y}\ln\big(1+e^{f_y(X)}\big)\right]\\
&\qquad\qquad\qquad\qquad\qquad\quad \text{(since } \ln(1+e^{-f_y(\mathbf{x})})\ge 0\ \forall(\mathbf{x},y))\\
&= \frac{1}{\ln(2)}\,\mathbf{E}_{(X,Y)\sim D}\big[\ell_{\text{OvA,log}}\big(Y,\mathbf{f}(X)\big)\big]\\
&= \frac{1}{\ln(2)}\,\text{er}_D^{\text{OvA,log}}[\mathbf{f}]\,.
\end{aligned}
$$

$\square$

## C    Proof of Theorem 3

*Proof.* Let $\mathbf{w}_1,\dots,\mathbf{w}_n \in \mathbb{R}^d$, $b_1,\dots,b_n \in \mathbb{R}$, and let $\mathbf{f}\in\mathcal{F}_{\text{spwlin}}$ be parametrized by $\{\mathbf{w}_y,b_y\}_{y=1}^n$, so that

$$f_y(\mathbf{x}) = \min_{y'\ne y}\big\{(\mathbf{w}_y - \mathbf{w}_{y'})^\top\mathbf{x} + (b_y - b_{y'})\big\}\quad \forall\mathbf{x}\,.$$

We will show that

$$\text{argmax}_{y \in [n]} f_y(\mathbf{x}) = \text{argmax}_{y \in [n]} \mathbf{w}_y^\top \mathbf{x} + b_y \,;$$

this will establish the result.

To see that the above claim is true, notice that we can write

$$f_y(\mathbf{x}) = (\mathbf{w}_y^\top \mathbf{x} + b_y) - \max_{y' \neq y} \left\{ \mathbf{w}_{y'}^\top \mathbf{x} + b_{y'} \right\}.$$

In other words, $f_y(\mathbf{x})$ is the difference between $(\mathbf{w}_y^\top \mathbf{x} + b_y)$ and the largest value of $(\mathbf{w}_{y'}^\top \mathbf{x} + b_{y'})$ among $y' \neq y$. Clearly, this difference is largest when $(\mathbf{w}_y^\top \mathbf{x} + b_y) \geq (\mathbf{w}_{y'}^\top \mathbf{x} + b_{y'}) \; \forall y' \neq y$ (in particular, in this case the difference is non-negative; in all other cases, the difference is negative, and therefore smaller). Thus

$$f_y(\mathbf{x}) \geq f_{y'}(\mathbf{x}) \; \forall y' \neq y \iff (\mathbf{w}_y^\top \mathbf{x} + b_y) \geq (\mathbf{w}_{y'}^\top \mathbf{x} + b_{y'}) \; \forall y' \neq y \,.$$

This proves the claim. □

## D   Proof of Corollary 4

This follows directly from the proof of Theorem 3.

## E   Details of Real Data Sets Used in Experiments in Section 5.2

Table 3: Multiclass classification data sets used in experiments in Section 5.2.

| Data set | # train | # validation | # test | # classes $(n)$ | # features $(d)$ |
|---|---|---|---|---|---|
| Covertype (50K) | 30000 | 10000 | 10000 | 7 | 54 |
| Digits | 5620 | 1874 | 3498 | 10 | 16 |
| USPS | 5468 | 1823 | 2007 | 10 | 256 |
| MNIST (70K) | 45000 | 15000 | 10000 | 10 | 780 |
| CIFAR10 | 37500 | 12500 | 10000 | 10 | 3072 |
| Sensorless | 35105 | 11702 | 11702 | 11 | 48 |
| Letter | 10500 | 4500 | 5000 | 26 | 16 |

**Notes:**

**Subsampling:** For Covertype, we used a random subsample of the original data set containing 50,000 examples (the original data set has 581,012 examples).

**Image data sets with pixel features:** The versions of the USPS and MNIST datasets that we used came with features scaled to the ranges $[-1, 1]$ and $[0, 1]$, respectively. For CIFAR10, we similarly scaled the features to the range $[0, 1]$ by dividing all features by 255.