[Reviews · NeurIPS 2020]

Review 1

Summary and Contributions: The paper proposes a fix for a group of multiclass classification losses, to make them calibrated surrogates for multiclass classification when the labeling function is linear. The fix is in essence a new family of classification losses that can be implemented easily with auto-differentiation libraries. The paper explains the issue, proposes a fix (backed by theoretical results), illustrates the theoretical results with an empirical study, and also shows that the family of losses introduced produce competitive results in datasets considered. I think the paper is clear and well executed.

Strengths: The paper does a great job isolating an issue, proposing a fix and backing the theoretical conclusion with empirical results. The most interesting to me is the conclusion that the new family of losses is competitive with existing losses and so gives more loss options for downstream applications. This work is relevant to members of the community with interest in Multiclass Classification.

Weaknesses: I only see minor issues (see feedback).

Correctness: The claims are correct.

Clarity: The paper is clearly written.

Relation to Prior Work: Yes.

Reproducibility: Yes

Additional Feedback: Thanks for the interesting and well-written paper. Broader impact: You could raise the point that multiclass classifiers are prone to perpetuating biases and being unfair [2], and so downstream applications should be attentive to these issues. For Definition 7, I would suggest citing Long & Servedio, but stating the definition as "A is B", rather than "Long & Servedio refer to A as B". Footnote 5: Is the actual definition identical or slightly more general? Another takeaway from Corollary 4 is that the algorithm there (with linear classifiers) is H-lin consistent, but not ERM. [220-221] if this margin is non-negative then $y \in \argmax$ [222] "Generalization to other multiclass models": You're starting to reach back to the CS-like losses, since the max can be moved into monotonic functions, and for this loss form Zhang 04 already had some consistency results (see eq 11). If there is interest in exploring generalization to other multiclass classification losses, Table 3 in [1] lists different losses in the literature, and many of them seem compatible with score functions of the form s'_i = s_i - max_{j \neq i} s_j. References [1] Multiclass Classification Calibration Functions, Pires & Szepesvari, 16. https://arxiv.org/pdf/1609.06385.pdf [2] A Survey on Bias and Fairness in Machine Learning, Mehrabi & al., 19. https://arxiv.org/pdf/1908.09635.pdf


Review 2

Summary and Contributions: The paper regards the notion of consistency (Bayes consistency and H-consistency) in multiclass classification. It examines the interplay between surrogate loss functions and a class of scoring functions that induce a classifier in defining consistency. In particular, the paper examines the conundrum put forth by Long and Servedio (2013) that minimizing convex calibrated surrogate loss functions over a class of linear scoring functions fails to recover the true classifier when the underlying data model is indeed linear. The authors resolve this seeming clash between Bayes consistency and H-consistency by considering a new class of ‘shared’ piecewise linear scoring functions for risk minimization and discuss a computational strategy for minimization over the more complex class. Further they present generalization of the idea to a given class of scoring functions by defining the class of ‘shared’ piecewise difference functions for improved classification accuracy. The beneficial effect of adopting this new class of scoring functions is demonstrated through numerical experiments.

Strengths: The paper brings clarity to statistical consistency properties in multiclass classification using surrogate loss functions. It illuminates intricate challenges of multiclass classification where surrogate loss functions, the class of scoring functions and the choice among joint/pairwise/one-vs-all approaches matter in defining consistency. The authors lay out the underlying issues very clearly and effectively, and present a clean remedy for the one-vs-all approach with a class of linear scoring functions by recognizing the fact that the decision regions given by arg max rule with component linear scoring functions are not linear but piecewise linear. Given that the one-vs-all approach is quite common in multiclass classification and linear scoring functions are often taken as a baseline function class, the proposed construction of a scoring function class that is more consistent with optimal classifiers has potential for improving classification accuracy in practice.

Weaknesses: While the authors articulate the reason for classification inconsistency with the class of linear scoring functions in one-vs-all approach and the need for defining the new class, I find the scope of the work somewhat limited, in part due to the very rigid notion of realizable H-consistency. The main idea is summed up in Figure 1 (very nicely), which leads to a clean solution for the very stylized noiseless setting. I wonder how the proposed approach could generalize well when there is label noise and one-vs-all boundaries are not necessarily of the same form, which is typically the case. The revelation that the partition given by arg max rule over linear component functions is piecewise linear rather than linear does not seem particularly profound. In addition, there is a trade-off between the simplicity in the formulation and resulting computation and consistency over a large class of data distributions. While the proposed approach could amend inconsistency in one-vs-all approach in some settings, the corresponding computation appears to require the same level of complexity as joint approaches (considering all classes simultaneously or “all-in-one”), which are known to be consistent for a broader class of distributions in general.

Correctness: The improvement in accuracy due to the new class of functions seems much more noticeable in simulation studies compared to the data applications. This qualitative difference might indirectly indicate the aforementioned gap between the highly stylized setting and more realistic settings. What might be the reason why the effect of a function class on accuracy is also quite different between logistic loss and hinge loss in Table 2? -- Post Rebuttal Update -- Regarding the numerical experiments, it would be good to have some thoughtful discussions as to the potential source of discrepancies between the two settings. Strong assumptions for the theory are probably not met in practice (e.g. rarely noiseless data, difference in the form of one-vs-rest boundaries), which limits the scope of the work. For the comment on the difference between hinge loss and logistic loss in terms of performance, I thought the presented theory and remedy in the paper do not necessarily differentiate them.

Clarity: It is very well written. One of the best (in terms of writing) among the papers I have reviewed this year!

Relation to Prior Work: I thought that the paper describes relation to other work clearly.

Reproducibility: Yes

Additional Feedback:


Review 3

Summary and Contributions: The authors are able to develop H-consistent algorithm from risk minimisation over a function class derived from linear scorers, which is proven to be H-consistent under the assumption that the target distribution is realizable. The H-consistency is proven and an implementation is evaluated on toy and real-world data.

Strengths: The relationships explored involving one-versus-all classification and the modified function class are interesting, and the theory of scoring rules, scoring functions and classifiers is a broad and deep topic in machine learning.

Weaknesses: The theory and derived function class and algorithm is an incremental contribution. The experimental results are encouraging, but do not strongly reinforce the theory.

Correctness: The empirical evaluation I believe could be improved. Since the main goal of the theory is to fix the issue of H-calibration (viz. Theorem 2), I would have appreciated more experiments or results which investigated the degree to which this is an issue, and how the new scoring class corrected for this. --- Post Rebuttal Update --- My concern raised was not a problem so much as something that I felt would help strengthen the paper. Thank you for drawing my attention to Table 2 and Figure 3. + Table 2 does not reinforce the issue of demonstrating the theoretical results on H-consistency (viz. Theorem 2) since the real world data sets are not obviously H_lin-realizable. The results here are mixed at best and do not give a clear picture. + Figure 3 (left) shows some benefit in low dimensions [2] with a low number of classes to predict [4], but in a higher dimensional setting [100] with a more moderate number of classes to predict [10] the benefit does not seem significant. To me this is not a very interesting result since high dimensional data is a cornerstone of machine learning and the experimental results appear to demonstrate that whatever problems may exist with the F_lin function class vanish in high dimensions.

Clarity: The terms realizable, consistent, and calibrated, I believe would benefit from a systematic definition early on. The current style of defining after use or as used makes it difficult to follow, and then finding a sequence of formal definitions half way through the paper is surprising. There is also quite a bit of redundancy in the definitions. • Bayes consistency is clearly a special case of H-consistency with the class H of measurable functions X → Y. • A similar redundancy is present at the top of page 5 where it would be simpler to introduce the generalisations of the error concepts up front, and explain how they map onto more conventional concepts. • The overloading of definitions in (6), (7) with (9), (10) when these are clearly almost identical concepts. Definitions 1 and 2: As stated in the introduction, the consistency concept is for an algorithm, but the notation, particularly in the two display equations, is misleading where ˆh is actually a sequence of random models obtained by evaluating a minimisation algorithm on the random samples S (presumably IID, but this is not necessarily specified?). The use of the union for the definition of the space of S is also confusing. Is it a mistake? It would make more sense for a data set of size m to live in the m-fold product of X × Y. --- Post Rebuttal Update --- I count seven theorem-like definitions in the exposition (not counting where the *same* concepts have been defined in the text), I believe this could be brought down to three or four, which would make more space for a richer story around the existing results. This is getting a little bit in the weeds, but if the data live in (X × Y)^m then then this would indeed allow the data to be of a variable size, provided m ≥ 1, as you desire, making the union redundant. (6), (9) missing "∀y∈Y"

Relation to Prior Work: The authors give a thorough account of the theoretical state of the art and how their work improves upon others.

Reproducibility: Yes

Additional Feedback: (1) The insight into H-consistency with respect to the refined function class is an interesting line of enquiry. I would have been interested to see more results along the line of Theorem 3. For example, what is the relationship between F_spwlin and F_lin? (2) The result Lemma 1 is actually something I have not seen before, can you prove it for a broader family of scorers? (3) The assumption that a distribution is H-realizable is very strong for application to real-world data. Can you prove any interesting results about the performance F_spwlin for prediction on non-H-realizable distributions? --- Post Rebuttal Update --- Thank you for drawing my attention to line 222 of the paper and answering (3) for me.


Review 4

Summary and Contributions: The manuscript points out an interesting observation about the apparent disagreement between two notions of consistency, in some settings, for the same calibrated and convex surrogate loss functions used in multi-class classification. The authors tease out this discrepancy observed in existing work, bridge the gap by proposing a suitable empirical minimization algorithm (for one-vs-all SVM) that ensures consistency, thereby improving the state-of-the-art. The contribution of this work is of interest and significance to the learning theoretic and ML communities, and the manuscript is clear and well-written. I advocate acceptance.

Strengths: The problem considered in the paper is important, and the questions considered are fundamental in nature, given the advances we have had in learning theoretic results since the seminal result of classification-calibration. The results are clearly written, the manuscript is fairly self-contained, and it is a pleasure to read.

Weaknesses: These are not weaknesses per se, but it would help to have some clarity on what appears to be a more general notion of H-consistency studied in this paper -- is there related work that studies identical definition (I see that the definition studied by Long and Servedio is less general)?

Correctness: Yes, I didn't verify the proofs carefully but.

Clarity: Yes. The paper is quite well-organized and well-written.

Relation to Prior Work: Yes

Reproducibility: Yes

Additional Feedback: I don't have much to add to the above feedback. Perhaps one question to the authors: What aspects of this work could be extrapolated to related supervised learning settings like, say, multi-label learning? ----- I've read the responses. I would very much want this paper to be accepted. ---

[Author Response · NeurIPS 2020]

**Author Response – Paper ID 12191 (Bayes Consistency vs. $\mathcal{H}$-Consistency)**

**Reviewer #1**

Thanks for all the suggestions (on broader impact, phrasing in Definition 7, and other improvements), and the additional references! We will incorporate all these suggestions and include these references in the final version if accepted.

Re. Footnote 5: The essence of the definition we have given (in Definition 7) is the same as that of Long and Servedio's definition, but technically, it is slightly more general in that we allow $\mathcal{F}$ to be any set of scoring function vectors $\mathbf{f} : \mathcal{X} \to \mathbb{R}^n$, while in the original definition $\mathcal{F}$ contained scoring function vectors $\mathbf{f} : \mathcal{X} \to \mathbb{R}^n$ whose component scoring functions $f_1, \ldots, f_n : \mathcal{X} \to \mathbb{R}$ all came from some fixed class of real-valued functions $\mathcal{F}_0 \subset \{f : \mathcal{X} \to \mathbb{R}\}$. We will modify the wording in Footnote 5 to clarify this.

**Reviewer #2**

Thanks for your comments – we are glad you enjoyed reading the paper!

Re. the realizable $\mathcal{H}$-consistency setting: We would like to clarify some aspects of this setting. (1) In general, we agree that $\mathcal{H}$-realizability can be a strong assumption on the distribution, and that universal Bayes consistency may be more desirable to achieve in practice. Nevertheless, we believe it is helpful to understand what is and what is not possible in the $\mathcal{H}$-realizable setting (part of the reason that there has been interest in this setting is that it is related to the classical PAC learning setting studied in computational learning theory). (2) It is true that there are other "all-in-one" approaches that are consistent for broader classes of distributions. Our goal here, however, is not to contrast one-vs-all algorithms with all-in-one algorithms; our main interest has been to examine the strengths of the various surrogate losses involved, and to emphasize that just because a particular surrogate loss does not produce expected results when minimized over a 'simple' scoring function class, it should not be immediately discarded or labeled ineffective, since it may be the case that it needs to be coupled with a different scoring function class in order to obtain the desired results. (3) Obtaining $\mathcal{H}$-consistent algorithms for general distributions (i.e. distributions that are not necessarily $\mathcal{H}$-realizable) is in general a computationally difficult problem. This is known from agnostic PAC learning theory; e.g. even finding an optimal (in 0-1 sense) *binary* linear classifier for non-linearly separable data is NP-hard. Note also that in the general non-$\mathcal{H}$-realizable/agnostic setting, $\mathcal{H}$-consistency becomes different from Bayes consistency.

Re. performance of hinge vs. logistic loss: For binary classification, the hinge loss comes with better regret transfer bounds (Bartlett et al., *JASA*, 2006), which could provide a partial explanation. It could be interesting to conduct a similar exploration in the multiclass case as well.

**Reviewer #3**

Re. experiments: Your question on how the new scoring class makes a difference is already answered in our experiments. In particular, in Figure 3 and Table 2, please compare the results for $\psi_{\mathrm{OvA,log/hinge}}$ with $\mathcal{F}_{\mathrm{lin}}$ and with $\mathcal{F}_{\mathrm{spwlin}}$.

Re. clarity: We are sorry that you found the ordering hard to follow. We have assumed some familiarity with the overall concepts in the introduction. We will try to re-order things somewhat in the final version if accepted. Note that the union $\cup_{m=1}^{\infty} (\mathcal{X} \times \mathcal{Y})^m$ simply allows the training sample to be of *any* size $m \geq 1$.

Re. additional comments: (1) $\mathcal{F}_{\mathrm{spwlin}}$ is a class of vectors of non-linear scoring functions with shared parameters; $\mathcal{F}_{\mathrm{lin}}$ is the class of vectors of linear scoring functions (see Figure 1, rows 2 and 3). There is no direct connection between them. (2) Lemma 1 is proved in the supplementary material (the proof is not hard, but to our knowledge the result is new). A similar result can be shown for more general function classes as well (see lines 222—228). (3) For non-$\mathcal{H}$-realizable distributions, achieving $\mathcal{H}$-consistency is generally NP-hard. Please see also our response to Reviewer #2 above.

**Reviewer #5**

Thanks for your comments – we are glad you liked the paper!

Re. notion of $\mathcal{H}$-consistency and relation to Long and Servedio's definition: Please see our response to Reviewer #1 above (comment on Footnote 5).

Re. other supervised learning settings such as multi-label learning: The question of consistency in general is certainly of interest in other supervised learning problems, and indeed there has been much work on understanding Bayes consistency for such problems in recent years (e.g. Duchi et al., ICML 2010; Gao & Zhou, COLT 2011; and many others in recent years). In all these cases, the target loss of interest is different from the 0-1 loss. For $\mathcal{H}$-consistency, when a distribution is $\mathcal{H}$-realizable (or simply realizable), it turns out the Bayes optimal model for all losses (that are zero on the diagonal and positive elsewhere) is the same as the Bayes optimal model for the 0-1 loss, and so one could in principle directly apply our results in such settings as well. But it could be interesting to consider other surrogate losses more commonly used for such problems, and other function classes $\mathcal{H}$ that may be more natural for them.

[Meta-Review · NeurIPS 2020]

Following the author response, there was broad consensus on the strengths of this submission: + studies a fundamental theoretical problem: while the issue of H realisable consistency is somewhat niche, furthering the understanding of the properties of surrogate losses is of clear interest, and progress in this space is welcome + the proposed fix to the inconsistency of common multiclass surrogates is intuitive, precisely studied theoretically, and demonstrated to be valuable empirically + the paper is very well-written and explained The discussion also revolved around some concerns, particularly raised by one reviewer: - mixed empirical results: in particular, the proposed fix does not consistently improve upon the original function class - potential limited scope of the theoretical results - potential redundancies with definitions and presentation The first point above was agreed on by other reviewers. However, their scores were unchanged as it was felt the results were sufficiently to furnish the theoretical results, which is the key point of the paper (as noted by the authors in Line 284). On my reading, I agree that the empirical results are sufficient for a primarily theoretical work. While there was no universal consensus on the second point above, on my reading I agree with the majority view of the other reviewers: the work appears a sufficiently self-contained contribution to an interesting problem, and the simplicity of the final solution is a salient feature. The resolution to the counter-intuitive findings of the original H realisable consistency paper are an important fundamental contribution. The third point above is minor, and somewhat at odds with the general positive scores other reviewers provided to the paper's readability. The authors are nonetheless encouraged to re-evaluate scope of trimming down some of the definitions. In summary, my recommendation is for this paper to be accepted.